# A Probiotic Mixture of *Lactobacillus rhamnosus* LR 32, *Bifidobacterium lactis* BL 04, and *Bifidobacterium longum* BB 536 Counteracts the Increase in Permeability Induced by the Mucosal Mediators of Irritable Bowel Syndrome by Acting on Zonula Occludens 1

**DOI:** 10.3390/ijms26062656

**Published:** 2025-03-15

**Authors:** Maria Raffaella Barbaro, Francesca Bianco, Cesare Cremon, Giovanni Marasco, Vincenzo Stanghellini, Giovanni Barbara

**Affiliations:** 1IRCCS Azienda Ospedaliero-Universitaria di Bologna, I-40138 Bologna, Italy; maria.barbaro2@unibo.it (M.R.B.); francesca.bianco@aosp.bo.it (F.B.); cesare.cremon@aosp.bo.it (C.C.); giovanni.marasco4@unibo.it (G.M.); v.stanghellini@unibo.it (V.S.); 2Department of Medical and Surgical Sciences (DIMEC), University of Bologna, I-40138 Bologna, Italy

**Keywords:** irritable bowel syndrome, permeability, probiotics, ZO-1

## Abstract

Irritable Bowel Syndrome (IBS) is a disorder of gut- brain interaction characterized by recurrent abdominal pain associated with altered bowel habits. The therapeutic options for IBS patients include the use of probiotics. The aim of this study was to assess the effect of a multi-strain probiotic made up by *Lactobacillus rhamnosus* LR 32, *Bifidobacterium lactis* BL 04, and *Bifidobacterium longum* BB 536 (Serobioma, Bromatech s.r.l., Milano, Italy) on an in vitro model of the intestinal epithelial barrier in the presence of mucosal mediators that are released by IBS patients. IBS (n = 28; IBS with predominant diarrhea, IBS-D = 10; IBS with predominant constipation, IBS-C = 9; and IBS with mixed bowel habits, IBS-M = 9) patients, diagnosed according to the Rome IV criteria, and asymptomatic controls (ACs, n = 7) were enrolled. Mucosal mediators that were spontaneously released by colonic biopsies were collected (supernatants). Two doses of Serobioma were tested with/without IBS/AC mediators. RNA was extracted from Caco-2 cells to evaluate the tight junction (TJ) expression. Serobioma (10^6^ CFU/mL) significantly reinforced the Caco-2 monolayer compared to growth medium alone (*p* < 0.05). IBS supernatants significantly increased Caco-2 paracellular permeability compared to the AC supernatants. The co-incubation of Caco-2 cells with IBS supernatants and Serobioma (10^6^ CFU/mL) avoided the paracellular permeability alterations that were induced by IBS supernatants alone (*p* < 0.001), and, in particular, IBS-D and IBS-M ones. The co-incubation of Serobioma (10^6^ CFU/mL) and IBS-D supernatants significantly increased ZO-1 expression compared to Caco-2 cells incubated with supernatants alone (*p* < 0.05), as confirmed via qPCR analyses. Serobioma (10^6^ CFU/mL) counteracts the paracellular permeability changes that are induced by IBS supernatants, in particular IBS-D and IBS-M supernatants, likely modulating ZO-1 expression.

## 1. Introduction

Irritable Bowel Syndrome (IBS) is a disorder of gut- brain interaction characterized by recurrent abdominal pain associated with altered bowel habits, including diarrhea (IBS-D), constipation (IBS-C), or a combination of both (IBS-M). A fourth subtype is the unclassified type (IBS-U). IBS affects approximately 10–15% of the global population and poses a significant healthcare burden due to the considerable impact on patients’ quality of life, the cumbersome diagnosis due to the absence of biomarkers, and the heterogeneity of the condition [1,2].

Although the precise pathophysiological mechanisms of IBS remain elusive, it is widely recognized as a multifactorial disorder involving a complex interplay of altered gut motility, visceral hypersensitivity, and psychosocial factors [3]. More recent research has highlighted the role of gut microbiota dysbiosis, intestinal permeability changes, and altered production of short-chain fatty acids (SCFAs) in IBS’s pathophysiology [4,5,6,7].

Increased intestinal permeability has garnered particular interest in the context of IBS, since it can allow for the translocation of microbial products into the lamina propria, which may trigger low-grade inflammation, and also into the bloodstream, contributing to IBS symptom development [7,8].

Given the multifaceted nature of IBS, a comprehensive approach to treatment is required, often combining dietary interventions, probiotics, pharmacotherapy, and psychological therapies to address the varied symptoms that are experienced by patients [9].

Probiotics are defined by the World Health Organization as “live microorganisms that, when administered in adequate amounts, confer a health benefit on the host” [10]. Among the proposed mechanisms of action of probiotics is gut barrier reinforcement [10], which can be achieved through several pathways: the competitive inhibition of pathogen adhesion, production of bioactive metabolites, stimulation of digestive enzymes, and synthesis of short-chain and branched-chain fatty acids [11,12].

*Lactobacilli* and *Bifidobacteria* are important components of the human gut microbiota and are commonly used as probiotics [13]. *B. longum* BB536 is a highly regarded probiotic bacterial strain with well-established benefits. Specifically, this strain plays a role in maintaining a balanced gut microbiota, can positively influence the immune response, and supports tight junction stability via exopolysaccharide and butyrate production [14,15,16]. A randomized, double-blind, placebo-controlled, and multicenter trial demonstrated that *B. longum* BB536, when added to standard treatment, improved symptoms in patients with mild to moderately active ulcerative colitis [17]. *L. rhamnosus* LR32 has shown immunomodulatory properties in vitro [18]. In an animal model, it helped restore the balance of the gut microbial community, enhanced the expression of tight junction proteins in both the ileum and hypothalamus, and supported the expression of genes involved in central 5-HT metabolism [19]. *B. lactis* BL04 is widely used in probiotic formulations with beneficial effects on gastrointestinal (GI) symptoms and on gut microbiota composition [20,21,22]. A randomized, double-blind, and placebo-controlled trial reported that *B. lactis* BL04 reduced the risk of upper respiratory tract infection in healthy subjects [23].

The aims of the present study were to characterize the effect of Serobioma, a multi-strain probiotic comprising *L. rhamnosus* LR32, *B. lactis* BL04, and *B. longum* BB 536, on an in vitro model of the intestinal epithelial barrier in the presence of mucosal mediators that are released by IBS patients compared to that of asymptomatic controls. In addition, molecular analyses were performed to evaluate the gene expression of zonula occludens-1 (ZO-1), occludin, and junctional adhesion molecule-A (JAM-A) to assess their possible involvement in permeability changes.

## 2. Results

### 2.1. Demographical Characteristics of Study Subjects

Table 1 shows the demographic characteristics of the study subjects. The IBS group was significantly younger than the AC group, and the same was the case for IBS-D and IBS-M patients. IBS-D patients were also significantly younger than IBS-C ones. A significantly higher percentage of women was present in the IBS-C group compared to the AC group (*p* < 0.05) and the other IBS subgroups (*p* = 0.05).

The IBS subtypes were comparable for abdominal pain and distension, except for the abdominal distention severity score, which was significantly higher in IBS-C patients compared to IBS-D ones.

The quality-of-life score was significantly higher in the AC group compared to IBS-M and IBS-D subtypes. In addition, IBS-D patients achieved the lowest quality of life score among the three IBS subtypes.

### 2.2. Assessment of pH and Cell Viability of Caco-2

In order to define two doses of Serobioma for further testing on Caco-2 cells, preliminary experiments were carried out by evaluating the effect of different doses on pH levels and Caco-2 viability. Thus, a change in these parameters could affect the permeability results.

As we have previously shown that a subset of IBS supernatants, containing the mucosal mediators that are spontaneously released by IBS biopsies, caused increased paracellular permeability in Caco-2 cells compared to supernatants from healthy individuals after 6 h of incubation [6], to choose the two doses of Serobioma, we focused on those that did not alter the pH or Caco-2 viability after 6 h of incubation. The highest dose of Serobioma that did not change these two parameters was 10^6^ CFU/mL, from now on referred to as S1 (Figure 1A–C). A second dose of 10^3^ CFU/mL, from now on referred to as S2, was selected arbitrary, and as shown in Figure 1, it did not influence the medium’s pH (Figure 1B) or the viability of the Caco-2 cells (Figure 1C).

### 2.3. Effect of Serobioma on Caco-2 Paracellular Permeability

Caco-2 cells were incubated with the medium alone or with the two concentrations of Serobioma, S1 and S2. Compared to the Caco-2 cells that were incubated with the medium alone, S1 significantly reinforced Caco-2 monolayer (*p* < 0.05, Figure 2).

### 2.4. Effect of IBS Supernatants on Caco-2 Paracellular Permeability

IBS supernatants caused a significant increase in paracellular permeability compared to the AC group after 6 h of incubation (*p* < 0.01). When analyzing IBS subtypes based on bowel habits, all subgroups showed a significant increase in Caco-2 paracellular permeability compared to the AC group (Figure 3).

### 2.5. Effect of Serobioma and IBS Supernatants on Caco-2 Paracellular Permeability

The effect of co-incubation of Serobioma and AC/IBS supernatants over time is shown in Figure 4. The highest dose of Serobioma (S1) significantly reduced the increase in paracellular permeability that was induced by the IBS supernatants (IBS vs. IBS+S1, *p* < 0.001). Regarding IBS subtypes, the highest dose of Serobioma significantly reduced the increase in paracellular permeability that was induced by the IBS-D (*p* < 0.01) and IBS-M (*p* < 0.01) supernatants, while no effect was observed with IBS-C ones.

### 2.6. Effect of Serobioma with/Without AC/IBS Supernatants on ZO-1, Occludin, and JAM-A Expression

To investigate the potential molecular mechanisms of Serobioma in reversing the paracellular permeability changes induced by IBS supernatants, qPCR analyses were performed on RNA extracted from Caco-2 cells at the end of the permeability experiments (Figure 5).The co-incubation of Serobioma and AC/IBS supernatants induced an increase in the expression of *ZO-1* compared to supernatants alone, although this difference only reached statistical significance for S1 that was co-incubated with IBS-D supernatants (Figure 5A). The highest dose of Serobioma also induced an increase in *occludin* expression with and without IBS supernatants, although statistical significance was not reached for any of the conditions (Figure 5B). The highest dose of Serobioma induced a significant increase in *JAM-A* expression compared to Caco-2 cells that were incubated with growth medium alone, while when co-incubated with IBS-D and IBS-M supernatants, it induced an increase in *JAM-A* expression which did not reach statistical significance (Figure 5C).

## 3. Discussion

In the present study, we demonstrated that 10^6^ CFU/mL of Serobioma prevents the paracellular permeability changes that are induced by IBS supernatants, particularly the IBS-D and IBS-M subtypes, on the Caco-2 monolayer after 6 h of incubation. In exploring the molecular mechanisms underlying this effect, we found that S1 induced an increase in the gene expression of *ZO-1* in Caco-2 cells that were incubated with IBS-D supernatants compared to supernatants alone.

The intestinal barrier is crucial for protecting the body from harmful antigens and maintaining homeostasis. Growing evidence indicates that the intestinal barrier is compromised, leading to increased permeability, in several intestinal and extra-intestinal conditions, including IBS [24]. The exact role of changes in the intestinal permeability in IBS remains debated, although it can be hypothesized that barrier disruption allows luminal antigens to activate the immune system, causing low-grade inflammation and altered neuro-immune interactions, which likely contribute to symptom development. Interestingly, among the therapeutic options for IBS patients, probiotics are noteworthy [25]. Certain strains of probiotics have been shown to restore barrier function by modulating TJ proteins and reducing permeability [26].

Here, we demonstrated that the use of Serobioma can avoid the paracellular permeability increase that is induced by mucosal mediators that are released by IBS biopsies, particularly, IBS-D. We also found that Serobioma induced an increase in *ZO-1* expression, suggesting that this could be one of the molecular mechanisms underlying the Serobioma effect. ZO-1 is a key protein involved in maintaining the integrity of TJs in the intestinal epithelium, playing a crucial role in regulating intestinal permeability. Using the same translational model based on Caco-2 cells and IBS mucosal mediators, previous evidence demonstrated that ZO-1 expression was reduced by IBS mediators [8,9]. Moreover, an altered expression or localization of ZO-1 has been reported in jejunal and colonic biopsies of IBS patients, suggesting an important role for this TJ protein in IBS [27,28].

Using the same probiotic formulation, previously published studies reported effects in the form of reinforcement of the Caco-2 monolayer by enhancing the expression of various TJ proteins, including ZO-1 [16]. In addition, the same research group demonstrated that in a co-culture model based on Caco-2 cells and HMC-1.2 (a mast cell line), Serobioma reduced cytokine production and barrier alterations, induced by LPS [29]. In our study, we used the mucosal mediators that are spontaneously released by IBS biopsies, which include several mast cell mediators (e.g., histamine, proteases, serotonin, cytokines, etc.), thus providing a stronger rationale for further evaluation of Serobioma in IBS [30].

Our study has some limitations: first, the use of an in vitro model is a highly simplified version of the intestinal environment; second, since this was a pilot study, only a limited number of samples were analyzed; third, there were differences in the age and percentage of females between the groups of subjects enrolled; fourth, we cannot exclude that diet may have a role in the composition of IBS supernatants and therefore in the results obtained, so we cannot generalize the obtained effect to geographical regions that are different from ours. However, this study has several important strengths: it utilized mediators that are spontaneously released from patient biopsies, which include molecules that are present in the gut; the preliminary pH tests ensured that the probiotic doses used during the chosen incubation period, did not alter the pH of the growth medium, thus avoiding a major error in the assessment of changes in permeability; finally, a control group of subjects was included to compare the results obtained in the IBS patient group.

Understanding the underlying mechanisms, particularly those related to the gut microbiota and intestinal paracellular permeability, remains an area of active research on the topic of IBS, with the aim of developing more targeted therapies. The results of this study provide an important starting point for future research aimed at clarifying the therapeutic effects of Serobioma in IBS.

## 4. Materials and Methods

### 4.1. Subjects

Consecutive patients with IBS (n = 28; IBS-D = 10; IBS-C = 9; and IBS-M = 9), diagnosed according to the Rome IV criteria, and asymptomatic controls (ACs, n = 7) were recruited at the Department of Medical and Surgical Sciences, University of Bologna (Italy). ACs were selected from subjects undergoing colonoscopy for colorectal cancer screening or following polypectomy and were individuals who met the exclusion criteria and had no gastrointestinal symptoms.

The exclusion criteria for all included subjects were as follows: any relevant organic, systemic, or metabolic disease; organic bowel diseases, including celiac disease or inflammatory bowel disease (Crohn’s disease, diverticular disease, ulcerative colitis, infectious colitis, ischemic colitis, microscopic colitis); previous major abdominal surgeries; patients consuming probiotics or receiving topical antibiotic therapy within the past month; and the use of corticosteroids, non-steroidal anti-inflammatory drugs, serotonergic agents, including 5-HT3 receptor antagonists (i.e., granisetron, ondansetron) and 5-HT4 receptor agonists (i.e., prucalopride), and tricyclic anti-depressants. In each subject, during the colonoscopy, four biopsies were taken at the level of the proximal descending colon to collect spontaneously released mediators.

All participants gave written informed consent. The protocol was approved by the Ethic Committee of Area Vasta Emilia Centro (CE-AVEC, approval identification number: 822/2021/Sper/AOUBo) and conducted in accordance with the Declaration of Helsinki.

### 4.2. Symptom Questionnaires

At the time of biopsy collection, patients completed a modified Italian version of the Bowel Disease Questionnaire (BDQ) to score the frequency and severity of their abdominal pain and bloating in the past 2 weeks prior to the interview [7].

Symptom severity was graded as 0–4 according to its impact on patients’ daily activities: 0—absent; 1—mild (not influencing activities); 2—relevant (diverting from, but not urging modification of, activities); 3—severe (influencing activities markedly enough to urge modifications); 4—extremely severe (precluding daily activities). The frequency was graded as 0–4 according to the following scale: 0—absent; 1—up to 1 day/week; 2—2 or 3 days/week; 3—4–6 days per week; 4—daily. The quality of life, as an expression of general well-being, was monitored using a 0–10 visual analog scale (0 = “I have never felt so bad”; 10 = “I have never felt so good”).

### 4.3. Mucosal Mediator Collection

Mucosal mediators spontaneously released from colonic biopsies, were collected according to a previously validated method [7]. Briefly, the biopsies were immediately immersed in plastic tubes containing 1 mL of Hepes–Krebs buffer solution (pH 7.4), weighed, and volume-adjusted to incubate 15 mg of biopsies in 1 mL of buffer. Incubation was carried out under continuous oxygenation at 37 °C for 25 min. Tubes were centrifuged at 200× *g* at 4 °C for 7 min, and supernatants containing mediators (called supernatants) were aliquoted and stored at −20 °C until the analysis. Before treatment, supernatants were filtered with 0.22 µm filters (Millex-GP, Millipore, Milan, Italy).

### 4.4. Caco-2 Cell Culture

A previously published protocol was followed to establish the in vitro model of the intestinal epithelial barrier based on the human intestinal epithelial cell line Caco-2 (EATCC, Port Down, UK) [7]. Briefly, cells were seeded onto porous filters (12-well Transwell Clear, 0.40 μm porosity, 1.1 cm of diameter; Corning, Milan, Italy) at a density of 200,000 cells/filter and cultured in Dulbecco’s modified Eagle’s medium (DMEM) (Gibco, Thermo Scientific, Milan, Italy), supplemented with 2 mmol/L L-glutamine (Gibco, Thermo Scientific, Milan, Italy), 50 IU/mL penicillin, 50 μg/mL streptomycin (Gibco, Thermo Scientific, Milan, Italy), and 10% heat-inactivated fetal bovine serum (FBS) (Gibco, Thermo Scientific, Milan, Italy) at 37 °C and using 5% of CO_2_. The culture medium was changed every 2 days. Using a volt-ohm-meter (Millicel^®^ ERS-2, Millipore, Milan, Italy), we measured the trans-epithelial cell resistance (TEER) every two days in order to follow the formation and differentiation of the Caco-2 monolayer.

### 4.5. Preparation of the Serobioma

A capsule containing a total of 8 × 10^9^ CFU living cells of *Lactobacillus rhamnosus* LR 32, *Bifidobacterium lactis* BL04, and *Bifidobacterium longum* BB 536, commercially available as Serobioma, was supplied by Bromatech S.r.l. (Milano, Italy). Serobioma was suspended in DMEM without supplements.

### 4.6. pH and Viability Assay

Caco-2 cells were seeded onto 24-well plates at a density of 70,000 CFU/mL and cultured until confluence. Before the assay, cells were washed with PBS and incubated with DMEM without supplements.

In the 30 h following incubation, using litmus/acid test papers, the change in pH was measured. In particular, the pH was assessed immediately after incubation and then every two hours for the first eight hours; subsequently, the pH was determined at the twenty-fourth and thirtieth hours.

Two concentrations of Serobioma, 10^6^ (S1) and 10^3^ CFU/mL (S2), that did not induce a pH change before 6 h of incubation, were selected to evaluate their effect on Caco-2 vitality.

A Sulforhodamine B colorimetric (SRB) assay was used to assess the cell viability. The method has been optimized to adherent cells in a 24-well plate. Cells were fixed with 50% trichloro acetic acid (TCA) at 4 °C for 1 h. Sequentially, the plate was washed 5 times with sterile water and left overnight to dry. The following day, cells were incubated with 300 μL of 0.4% SRB, dissolved in 1% acetic acid, for 30 min in the dark; then, the plate was washed 4 times with 200 μL of 1% acetic acid. After solubilization with TRIS 10 mM pH 10.5, the absorbance was measured at λ = 540 nm using a spectrophotometer (TECAN Spark, Milan, Italy). The absorbance was directly proportional to the protein content and therefore to the number of live cells that were present in each well. Caco-2 viability was assessed after 6 h of incubation with the two concentrations of Serobioma or only with DMEM served as a blank (control). The pH and cell viability were assessed for each of the above-mentioned conditions, at least in duplicate.

### 4.7. Permeability Assay

Paracellular permeability alterations were assessed by evaluating the flow of Fluorescein-5-(and-6)-Sulfonic Acid (FITC 0.1 mg/mL; absorption/emission peak: 485/535, Invitrogen) through the Caco-2 monolayer over time. A standard curve was used to convert the absorbance results into concentrations of FITC.

Basolateral aliquots of 150 μL were taken, and the amount of FITC was measured using a 96-well fluorescent plate reader (Spark Multimode Microplate Reader, TECAN, Milan, Italy). After the measurement, the aliquots were re-added to the basolateral side. Absorbance evaluations were performed at time 0 (immediately after the addition of treatments), every 30 min for the first 2 h, and then each hour for the subsequent 4 h. The FITC absorbance values obtained were directly proportional to the permeability changes in the cell monolayer.

### 4.8. CaCo-2 Treatments and Permeability Evaluation

Caco-2 cell monolayers were incubated for 6 h with supernatants of ACs and IBS patients with or without the simultaneous incubation of S1 and S2 of Serobioma. In case of co-incubation, Serobioma was added to Caco-2 cells at the same time as the supernatants. Each condition was tested in duplicate.

### 4.9. qPCR Analyses

After the paracellular permeability experiments, the Caco-2 cells were rinsed with PBS and preserved in RNAlater buffer until RNA extraction (RNeasy Minikit, Qiagen, Milan, Italy) and reverse transcription using the SuperScript™VILO™ cDNA Synthesis Kit in a final volume of 20 µL, according to the manufacturer’s instructions (Thermo Scientific, Milan, Italy). The gene expression was assessed via qPCR (PowerUp SYBR Green, Thermo Scientific, Milan, Italy) for *ZO-1*, *occludin*, and *JAM-A* on a QuantStudio Real-Time PCR Systems 5 (Thermo Scientific, Milan, Italy). The expression of each gene was normalized to the reference gene *Glyceraldehyde-3-phosphate dehydrogenase* (*GAPDH*). 

The following primers were used: *ZO-1*: forward 5′-gaatgatggttggtatggtgcg-3′, reverse 5′-tcagaagtgtgtctactgtccg-3′; occluding forward 5′-tcctataaatccacgccggttc-3′, reverse 5′-ctcaaagttaccaccgctgctg-3′; JAM-A forward 5′-cagaggtgattcatggctctgtg-3′, reverse 5′-ttccaggctggcaataactgac-3′; *GAPDH*: forward 5′-cagcaagagcacaagaggaag-3′, reverse 5′-caactgtgaggaggggagatt-3′.

The amplification conditions were as follows:*ZO-1* gene: 15 min at 95 °C, followed by 40 cycles of 15 s at 95 °C, 30 s at 53 °C, and 30 s at 72 °C;*Occludin* gene: 15 min at 95 °C, followed by 40 cycles of 15 s at 95 °C, 30 s at 53 °C, and 30 s at 72 °C;*JAM-A* gene: 15 min at 95 °C, followed by 40 cycles of 15 s at 95 °C, 30 s at 53 °C, and 30 s at 72 °C*GAPDH* gene: 15 min at 95 °C, followed by 40 cycles of 15 s at 95 °C, 30 s at 53 °C, and 30 s at 72 °C.

The melting curve data were analyzed at the end of each reaction. The relative gene expression was calculated using the corresponding condition without Serobioma as the calibrator. A negative control for the PCR reaction (1 µL of water instead of cDNA) and a no-reverse-transcription control were included on each PCR plate. Each reaction was performed in duplicate, and the mean threshold cycle (Ct) was determined from the two runs. The relative gene expression was calculated using the ΔΔCt method, and the gene expression was reported as the fold difference (2^−ΔΔCt^).

### 4.10. Statistical Analysis

Results are expressed as mean ± SE. Statistical analysis was carried out with the computer-assisted PrismGraphPad program (Prism version 8.1; GraphPad Software, SanDiego, CA, USA). Kruskal–Wallis test was used for multiple comparison; Mann–Whitney U-test was used for paired comparisons; Spearman rank test was used for correlation analysis; and Pearson χ2 was used for gender analysis. *p* values < 0.05 were considered significant.

## Figures and Tables

**Figure 1 ijms-26-02656-f001:**
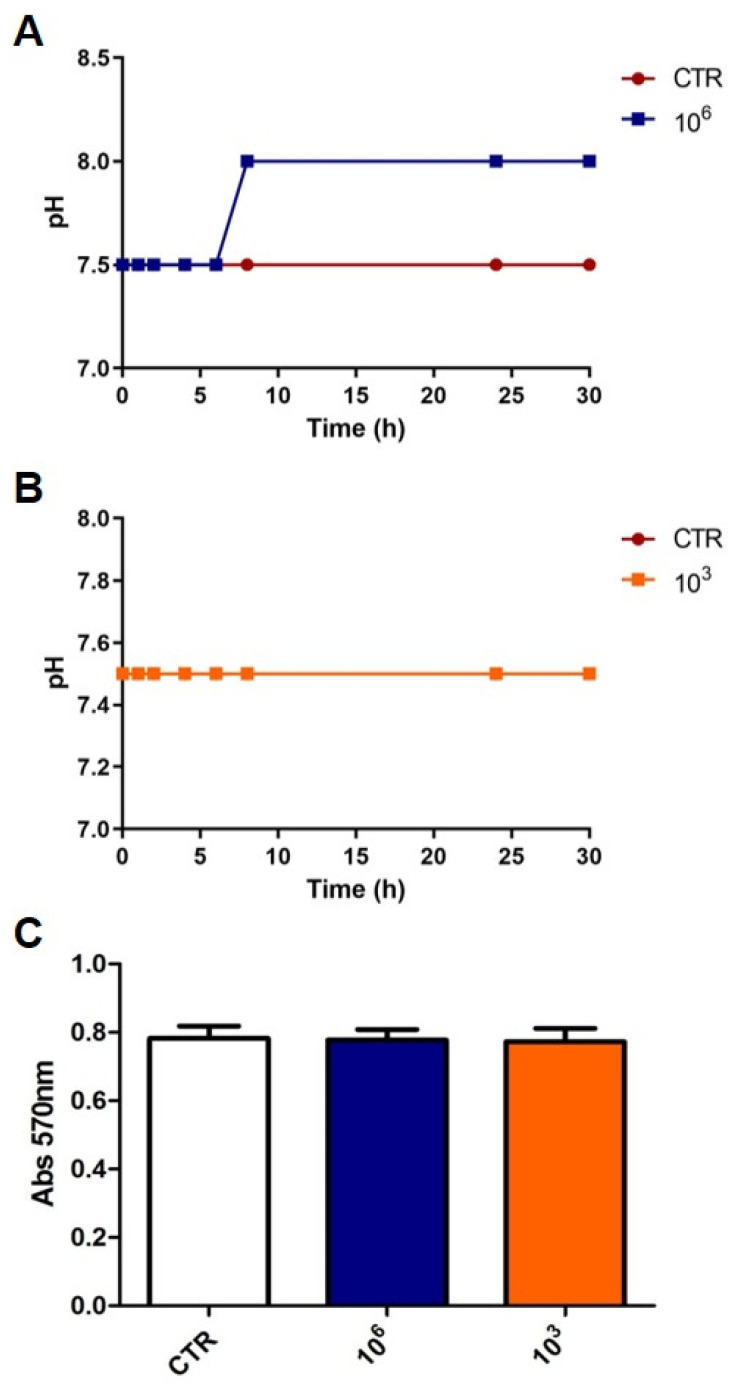
Effect of Serobioma on pH of medium and Caco-2 viability. (**A**,**B**) pH evaluation after 2, 4, 6, 8, 24, and 30 h of cell incubation with or without Serobioma (10^6^ CFU/mL and 10^3^ CFU/mL, respectively). (**C**) Caco-2 viability assay after 6 h of incubation with or without Serobioma (10^6^ and 10^3^ CFU/mL). CTR: Caco-2 cells incubated with medium alone (n = 4); 10^6^: Caco-2 cells incubated with 10^6^ CFU/mL (n = 2); 10^3^: Caco-2 cells incubated with 10^3^ CFU/mL (n = 2). Each condition was tested in triplicate.

**Figure 2 ijms-26-02656-f002:**
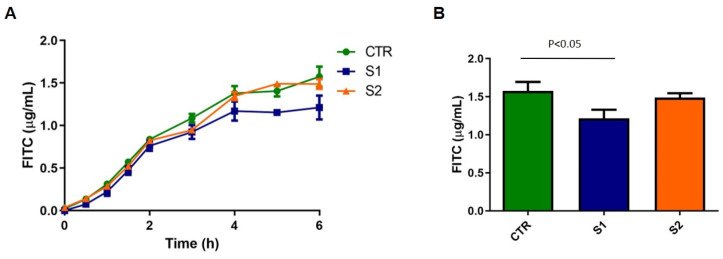
Effect of Serobioma on Caco-2 paracellular permeability. (**A**) Paracellular permeability changes during 6 h of Caco-2 incubation with/without Serobioma. (**B**) Paracellular permeability changes at 6 h of Caco-2 incubation with/without Serobioma. CTR: Caco-2 cells incubated with medium alone (n = 7); S1: Caco-2 cells incubated with 10^6^ CFU/mL (n = 6); S2: Caco-2 cells incubated with 10^3^ CFU/mL (n = 4).

**Figure 3 ijms-26-02656-f003:**
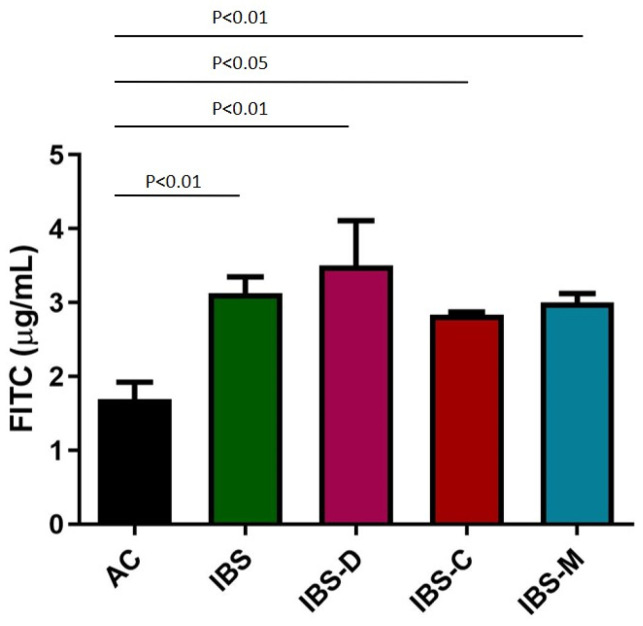
Effect of AC/IBS supernatants on Caco-2 paracellular permeability after 6 h of incubation. AC: Caco-2 cells incubated with AC supernatants (n = 7); IBS: Caco-2 cells incubated with IBS-D supernatants (n = 10) plus Caco-2 cells incubated with IBS-C supernatants (n = 9) plus Caco-2 cells incubated with IBS-M supernatants (n = 9); IBS-D: Caco-2 cells incubated with IBS-D supernatants (n = 10); IBS-C: Caco-2 cells incubated with IBS-C supernatants (n = 9); and IBS-M: Caco-2 cells incubated with IBS-M supernatants (n = 9).

**Figure 4 ijms-26-02656-f004:**
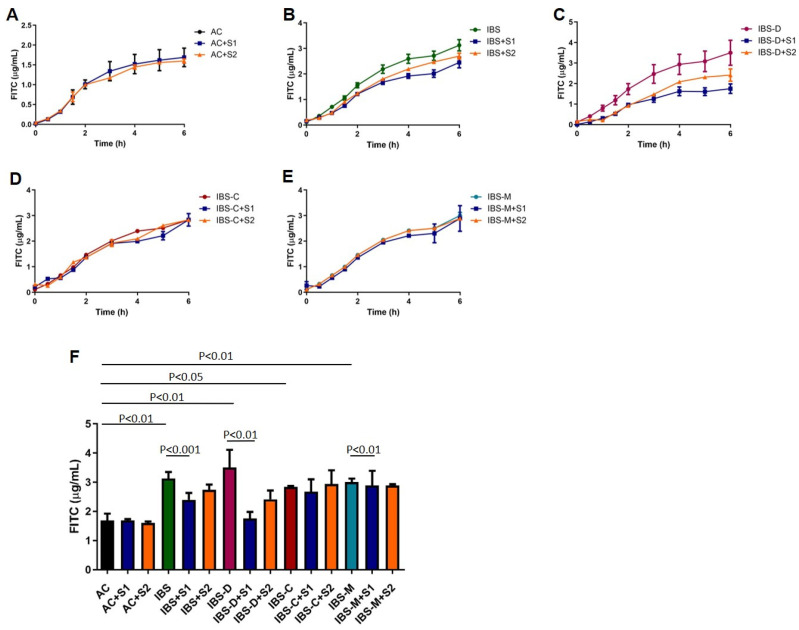
Effect of Serobioma being co-incubated with AC/IBS supernatants. (**A**) Effect of co-incubation of AC and Serobioma over time; (**B**) effect of co-incubation of IBS (IBS-D plus IBS-C plus IBS-M) and Serobioma over time; (**C**) effect of co-incubation of IBS-D and Serobioma over time; (**D**) effect of co-incubation of IBS-C and Serobioma over time; (**E**) effect of co-incubation of IBS-M and Serobioma over time; (**F**) depiction of paracellular permeability changes after 6 h of incubation. AC: Caco-2 cells incubated with AC supernatants; AC+S1: Caco-2 cells incubated with AC supernatants and 10^6^ CFU/mL of Serobioma; AC+S2: Caco-2 cells incubated with AC supernatants and 10^3^ CFU/mL of Serobioma; IBS: Caco-2 cells incubated with IBS-D supernatants plus Caco-2 cells incubated with IBS-C supernatants plus Caco-2 cells incubated with IBS-M supernatants; IBS+S1: Caco-2 cells incubated with IBS supernatants and 10^6^ cell/well Serobioma; IBS+S2: Caco-2 cells incubated with IBS supernatants and 10^3^ CFU/mL of Serobioma; IBS-D: Caco-2 cells incubated with IBS-D supernatants; IBS-D+S1: Caco-2 cells incubated with IBS-D supernatants and 10^6^ CFU/mL of Serobioma; IBS-D+S2: Caco-2 cells incubated with IBS-D supernatants and 10^3^ CFU/mL of Serobioma; IBS-C: Caco-2 cells incubated with IBS-C supernatants; IBS-C+S1: Caco-2 cells incubated with IBS-C supernatants and 10^6^ CFU/mL of Serobioma; IBS-C+S2: Caco-2 cells incubated with IBS-C supernatants and 10^3^ CFU/mL of Serobioma; IBS-M: Caco-2 cells incubated with IBS-M supernatants; IBS-M+S1: Caco-2 cells incubated with IBS-M supernatants and 10^6^ CFU/mL of Serobioma; and IBS-M+S2: Caco-2 cells incubated with IBS-M supernatants and 10^3^ CFU/mL of Serobioma.

**Figure 5 ijms-26-02656-f005:**
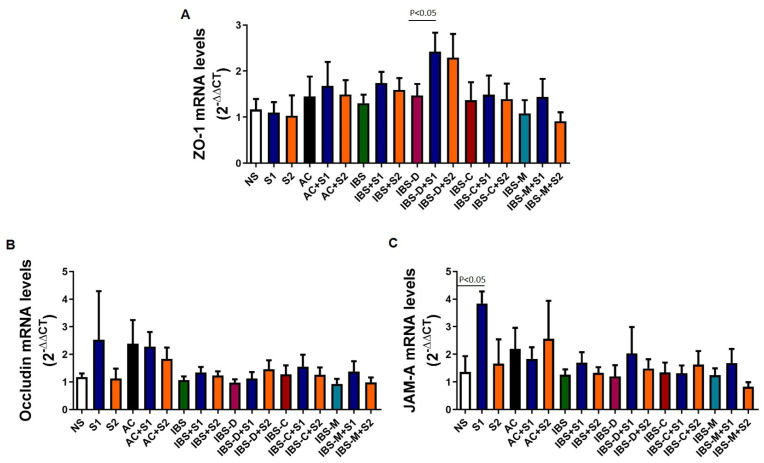
Effect of Serobioma with/without AC/IBS supernatants on *ZO-1* (**A**), *occludin* (**B**), and *JAM-A* (**C**) expression. NS: Caco-2 cells incubated with medium alone; S1: Caco-2 cells incubated with 10^6^ CFU/mL of Serobioma; S2: Caco-2 cells incubated with 10^3^ CFU/mL of Serobioma. AC: Caco-2 cells incubated with AC supernatants; AC+S1: Caco-2 cells incubated with AC supernatants and 10^6^ CFU/mL of Serobioma; AC+S2: Caco-2 cells incubated with AC supernatants and 10^3^ CFU/mL of Serobioma; IBS: Caco-2 cells incubated with IBS-D supernatants plus Caco-2 cells incubated with IBS-C supernatants plus Caco-2 cells incubated with IBS-M supernatants; IBS+S1: Caco-2 cells incubated with IBS supernatants and 10^6^ cell/well Serobioma; IBS+S2: Caco-2 cells incubated with IBS supernatants and 10^3^ CFU/mL of Serobioma; IBS-D: Caco-2 cells incubated with IBS-D supernatants; IBS-D+S1: Caco-2 cells incubated with IBS-D supernatants and 10^6^ CFU/mL of Serobioma; IBS-D+S2: Caco-2 cells incubated with IBS-D supernatants and 10^3^ CFU/mL of Serobioma; IBS-C: Caco-2 cells incubated with IBS-C supernatants; IBS-C+S1: Caco-2 cells incubated with IBS-C supernatants and 10^6^ CFU/mL of Serobioma; IBS-C+S2: Caco-2 cells incubated with IBS-C supernatants and 10^3^ CFU/mL of Serobioma; IBS-M: Caco-2 cells incubated with IBS-M supernatants; IBS-M+S1: Caco-2 cells incubated with IBS-M supernatants and 10^6^ CFU/mL of Serobioma; and IBS-M+S2: Caco-2 cells incubated with IBS-M supernatants and 10^3^ CFU/mL of Serobioma. Expression was calculated considering corresponding condition without Serobioma as calibrator (e.g., the expression of “IBS-D+S1” was calculated using “IBS-D” as calibrator).

**Table 1 ijms-26-02656-t001:** Demographic and clinical characteristics of study subjects.

	AC (n = 7)	IBS (n = 28)	IBS-D (n = 10)	IBS-C (n = 9)	IBS-M (n = 9)	*p* Values
Age	56 ± 4.8	41 ± 2.4 ^b^	33 ± 2.1 ^c^	50 ± 4.4 ^d^	40 ± 4.2 ^b^	*p* < 0.01 ^a^
Percentage of women	2 (29%) ^g^	16 (57%)	4 (40%) ^g^	8 (89%)	4 (44%) ^g^	0.065
Abdominal pain severity	-	1.7 ± 0.2	2 ± 0.5	1.4 ± 0.5	1.6 ± 0.4	0.627
Abdominal pain frequency	-	1.8 ± 0.3	2.3 ± 0.5	1.3 ± 0.6	1.9 ± 0.4	0.388
Abdominal distention severity	-	1.7 ± 0.2	1.4 ± 0.4	2.3 ± 0.2 ^e^	1.6 ± 0.2	0.078
Abdominal distention frequency	-	2.4 ± 0.3	1.9 ± 0.6	3.1 ± 0.3	2.3 ± 0.4	0.203
Quality of life	7.9 ± 0.3	4.8 ± 0.5 ^f^	3.1 ± 0.7 ^e,f^	5.5 ± 1.1	5.7 ± 0.5 ^f^	*p* < 0.01 ^a^

Data are shown as mean ± SE, median and interquartile range [in brackets], or absolute and relative frequencies. ^a^: Kruskal–Wallis test (HC vs. IBS-D vs. IBS-C vs. IBS-M or IBS-D vs. IBS-C vs. IBS-M) for abdominal pain and distension or Pearson χ^2^ (for female percentage); ^b^: *p* < 0.05 vs. AC (Mann–Whitney test); ^c^: *p* < 0.001 vs. AC (Mann–Whitney test); ^d^: *p* < 0.01 vs. IBS-D (Mann–Whitney test); ^e^: *p* < 0.05 vs. IBS-M (Mann–Whitney test); ^f^: *p* < 0.01 vs. AC (Mann–Whitney test); ^g^: *p* < 0.05 vs. IBS-C (Pearson χ^2^).

## Data Availability

The data presented in this study are available in Zenodo at the following link: https://doi.org/10.5281/zenodo.14826087.

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
