# Peer review of "A Probiotic Mixture of Lactobacillus rhamnosus LR 32, Bifidobacterium lactis BL 04, and Bifidobacterium longum BB 536 Counteracts the Increase in Permeability Induced by the Mucosal Mediators of Irritable Bowel Syndrome by Acting on Zonula Occludens 1"

_ijms, 2025, doi:10.3390/ijms26062656_

Round 1

Reviewer 1 Report

Comments and Suggestions for Authors

REVIEW

Dear authors,

The work proposes the use of a mixture of commercial probiotics (Serobioma) in the regulation of cell permeability in patients with irritable bowel syndrome, using an in vitro model. The increase in cell permeability allows the translocation of both microorganisms and their metabolites, which frequently causes endotoxemia in patients and systemic inflammatory processes, so using this type of therapy can have a beneficial impact on the reduction of symptoms.

Please amend the requested comments and submit the revision file.

  1. I believe that when evaluating a commercial probiotic, they should replace the names of the 3 microorganisms with that of the product (Serobioma), as well as add the word “in vitro model”, since mentioning the pathology of irritable bowel syndrome is related to patients.

  1. In the Abstract section you must write the full name of the microorganisms since they are abbreviated in the genus, as well as write them in italics.

  1. Section Introduction lacks information regarding the commercial probiotic and evidence of the health benefits that have been attributed to the 3 microorganisms that compose it. Add references regarding the commercial product Serobioma.

  1. Table 1. Review the percentage of the IBS group column and female sex (57800%).

  1. Although they mention that the limitations of the work were the small number of biopsies available from the patients, age is a very important factor as well as the diet of the patients, the Mediterranean diet is very different from the Eastern or Latin diet, to name two geographic regions, which means that they could not generalize the effect obtained. They should consider the animal model for a closer approach to the results.

  1. They did not evaluate a very important aspect, which is the immunological aspect in Caco-2 cells; the large quantity and diversity of metabolites in the patient samples could trigger different responses upon contact. I believe that the in vitro model used should be complemented.

  1. They must review and correct the correct writing of units such as “mL” and “mL”.

  1. How they selected the working doses of 1x103 y 1x106 CFU/mL?

  1. I believe that the addition of fluorescence microscopy images would provide visual evidence of permeability regulation by probiotic microorganisms.

  1. Section is missing Conclusions.

Please amend the requested comments and submit the revision file.

Author Response

We are grateful to the editors for giving us the opportunity to respond to the questions raised during the revision process. We have carefully considered all the issues and we have clarified some areas helpful for the readers in the submitted new version of our manuscript.

Comments from the Reviewers:

Reviewer: 1

Dear authors,

The work proposes the use of a mixture of commercial probiotics (Serobioma) in the regulation of cell permeability in patients with irritable bowel syndrome, using an in vitro model. The increase in cell permeability allows the translocation of both microorganisms and their metabolites, which frequently causes endotoxemia in patients and systemic inflammatory processes, so using this type of therapy can have a beneficial impact on the reduction of symptoms.

We thank the reviewer for his/her comments and suggestions.

  1. I believe that when evaluating a commercial probiotic, they should replace the names of the 3 microorganisms with that of the product (Serobioma), as well as add the word “in vitromodel”, since mentioning the pathology of irritable bowel syndrome is related to patients. 

Response: We thank the reviewer for these suggestions. We have changed the manuscript accordingly.

  1. In the Abstractsection you must write the full name of the microorganisms since they are abbreviated in the genus, as well as write them in italics. 

Response: We thank the reviewer and have changed the abstract accordingly.

  1. Section Introductionlacks information regarding the commercial probiotic and evidence of the health benefits that have been attributed to the 3 microorganisms that compose it. Add references regarding the commercial product Serobioma. 

Response: We have added new references to the introduction to include the information suggested by the reviewer.

  1. Table 1.Review the percentage of the IBS group column and female sex (57800%).

Response: We thank the reviewer for highlighting this mistake. We have corrected it accordingly.

  1. Although they mention that the limitations of the work were the small number of biopsies available from the patients, age is a very important factor as well as the diet of the patients, the Mediterranean diet is very different from the Eastern or Latin diet, to name two geographic regions, which means that they could not generalize the effect obtained. They should consider the animal model for a closer approach to the results. 

Response: We agree with the reviewer on the importance of age and diet, as well as other factors related to the geographical origin of the subjects studied, which could therefore lead to different results. We have included this concept in the discussion. The control subjects are mainly recruited from patients undergoing colonoscopy for colorectal cancer screening or for follow-up after polypectomy, so their average age is around 50 years, and unfortunately this pool for recruitment of controls does not always match the age of IBS patients.

We agree that we used an in vitro model, which, like all in vitro models, has some limitations, as all in vitro models, but there are some important points to consider. We used a reverse-translational model based on Caco-2 cells incubated with mediators spontaneously released by IBS colonic biopsies, and not the “simple Caco-2 monolayer model”. This model is tightly controlled by measuring TEER and by administering IBS mediators at precise doses and times. This model is widely used in the literature and our group has published several papers using it in the past (PMID: 18824556, PMID: 29956419). There are some limitations to replicating the experiments in other models, such as animal models. The first is that many mediators are no longer available. In addition, the setup and permeability experiments would take several months. Finally, there isn’t a single animal model for IBS, but several models that mimic different aspects of IBS. Although interesting, this work represents a whole new project and is beyond the scope of this paper.

  1. They did not evaluate a very important aspect, which is the immunological aspect in Caco-2 cells; the large quantity and diversity of metabolites in the patient samples could trigger different responses upon contact. I believe that the in vitro model used should be complemented. 

Response: We thank the reviewer for this comment highlighting the ability of the Caco2 cell line to produce inflammatory mediators. We agree with the reviewer that stimulation of Caco2 with mediators can induce an immune response. There are some previously published data on the effect of Serobioma on cytokine production in vitro (PMID: 36835568, PMID: 29674267). However, the present study aimed to assess whether Serobioma was able to reverse the increased permeability induced by mediators spontaneously released from biopsies of patients with IBS. It is likely that the mediators induce an immunological response and that this contributes to the observed increase in permeability. Similarly, we cannot exclude the possibility that the recovery of permeability induced by incubation with the probiotic mixture occurs, at least in part, through mechanisms induced by the immune response. Although it would be interesting to evaluate the secretion of inflammatory cytokines released by Caco-2, the mediators are no longer available. A new study would be required to correlate these results with those of permeability.

  1. They must review and correct the correct writing of units such as “mL” and “mL”. 

Response: We checked the manuscript accordingly to the reviewer’s comment.

  1. How they selected the working doses of 1x103y 1x106 CFU/mL? 

Response: The two doses were selected on the basis of ancillary experiments in which we incubated different doses of Serobioma with Caco-2 cells and assessed the pH of the growth medium. pH is a factor that influences permeability, so we monitored it over time. We focused on the doses that did not change the pH after 6 hours of incubation because we had previously reported an effect of IBS mediators on Caco-2 permeability at this time point, compared to control supernatants. 106 was the highest dose that did not change the pH after 6 h. Therefore, 106 and lower doses were incubated with Caco-2 to assess their viability. We found 4 doses that did not induce a change in Caco-2 viability: 10^6. 105, 104, and 103 (we did not consider a dilution below 103 because it would be extremely diluted). 106 and 103 were chosen as the most different to avoid testing two very similar doses. We have better explained this point in the materials and methods section of the revised version manuscript.

  1. I believe that the addition of fluorescence microscopy images would provide visual evidence of permeability regulation by probiotic microorganisms.

Response: We thank the reviewer for this suggestion, which we agree would be interesting. Unfortunately, most of the supernatants are no longer available to perform these experiments. We consider this an important part to be developed in future studies.

  1. Section is missing Conclusions.

Response: We had not included the conclusions section because it is reported as optional in the instructions for authors, and only recommended if the discussion is particularly long or complex.

Reviewer 2 Report

Comments and Suggestions for Authors

The manuscript of Barbaro et al. concerns variations of irritable bowel syndrome. In an in vitro model the effects of supernatants from biopsies of patients on monolayers of Caco-2 cells are tested and by adding a mixture of probiotic bacteria the effects are tried to be inhibited. This is interesting. However, the manuscript is apparently very hastily written. Often, what appears to me as lab jargon is used. This results in some statements which are simply wrong. Thus, the manuscript should be rejected and the authors should be ask to resubmit the manuscript after critically editing it. Many of such irritating statements are found throughout the manuscript. I will point out only a few.

  1. Abstract: IBS-D..etc. does not make sense here unless explained.
  2. Abstract: Caco-2 permeability would mean that compounds can reach the inside of Caco cells. This is nonsense. Paracellular permeability of Caco-2 monolayers is meant. This is found through out the manuscript and should be corrected.
  3. English in general is ok but at some places a native speaker could improve the style.
  4. Introduction: improvement of intestinal permeability ….this means that permeability is increased but the contrary is meant. Please correct.
  5. Introduction: the authors claim that that they identified the underlying molecular mechanism. By testing three proteins involved in tight junctions? This is certainly a vast overstatement. Please, rephrase.
  6. Results,2.2: this section should start with a description of the experimental design and how they were performed, including to mention monolayers, and later transwell cultures. Also IBS supernatants precisely should be described more precisely. At least once IBS biopsy supernatant needs to be mentioned. If this is supposed to be an abbreviation, it needs to be introduced.
  7. How often were the experiments carried out. The number of samples are given but not the reproducibility of the results.
  8. Fig.1: which assay is used for viability. Not described here or in M&M. To give here OD does not make sense therefore.
  9. Results, 2,3: …recovered paracellular permeability… is wrong. Since bacteria and supernatant are given at the same time the effect is avoided or inhibited since it is not induced at the first place. This statement is also found at other places.
  10. Results: paragraph 2.3 is found twice.
  11. How is concentration of FITC calculated?
  12. Why was FITC diffusion used instead of TEER.
  13. Fig. 5: what was measured should be shown in the panels.
Comments on the Quality of English Language

see above

Author Response

Reviewer: 2

The manuscript of Barbaro et al. concerns variations of irritable bowel syndrome. In an in vitro model the effects of supernatants from biopsies of patients on monolayers of Caco-2 cells are tested and by adding a mixture of probiotic bacteria the effects are tried to be inhibited. This is interesting. However, the manuscript is apparently very hastily written. Often, what appears to me as lab jargon is used. This results in some statements which are simply wrong. Thus, the manuscript should be rejected and the authors should be ask to resubmit the manuscript after critically editing it. Many of such irritating statements are found throughout the manuscript. I will point out only a few.

We thank the reviewer for finding the experimental design of our study interesting. We have reread the manuscript very carefully and have corrected the inconsistencies pointed out by the reviewer and others we have identified.

  1. Abstract: IBS-D..etc. does not make sense here unless explained.

Response: The spelling for IBS-D, IBS-C and IBS-M has been introduced in the abstract.

  1. Abstract: Caco-2 permeability would mean that compounds can reach the inside of Caco cells. This is nonsense. Paracellular permeability of Caco-2 monolayers is meant. This is found through out the manuscript and should be corrected.

Response: We have changed Caco-2 permeability to Caco-2 paracellular permeability throughout the manuscript.

  1. English in general is ok but at some places a native speaker could improve the style.

Response: We thank the reviewer for this suggestion and have improved some parts of the manuscript

  1. Introduction: improvement of intestinal permeability ….this means that permeability is increased but the contrary is meant. Please correct.

Response: We have changed the sentence to “gut barrier reinforcement” as reported by Hill, C.; Guarner, F.; Reid, G.; Gibson, G.R.; Merenstein, D.J.; Pot, B.; Morelli, L.; Canani, R.B.; Flint, H.J.; Salminen, S.; et al. The International Scientific Association for Probiotics and Prebiotics Consensus Statement on the Scope and Appropriate Use of the Term Probiotic. Nat Rev Gastroenterol Hepatol 2014, 11, 506–514, doi:10.1038/nrgastro.2014.66.

  1. Introduction: the authors claim that that they identified the underlying molecular mechanism. By testing three proteins involved in tight junctions? This is certainly a vast overstatement. Please, rephrase.

Response: We have changed the sentence accordingly to the reviewer’s comment.

  1. Results,2.2: this section should start with a description of the experimental design and how they were performed, including to mention monolayers, and later transwell cultures. Also IBS supernatants precisely should be described more precisely. At least once IBS biopsy supernatant needs to be mentioned. If this is supposed to be an abbreviation, it needs to be introduced.

Response: We clarified this point in the revised version of the manuscript.

  1. How often were the experiments carried out. The number of samples are given but not the reproducibility of the results.

Response: In each permeability experiment (i.e. in each plate), the control condition (Caco-2 cells incubated with growth medium alone) was present. In the same way, the two doses of Serobioma (106 and 103) were tested in different plates. The results were comparable among different plates, as shown by the standard errors reported in figure 1. The other conditions, i.e. the supernatants alone and the supernatants + two doses of the probiotic mixture were not tested in different plates (i.e. in independent experiments) because the volume of the supernatants was insufficient.  It should be remembered that each supernatant was used in the following conditions: alone and in the presence of the two doses of probiotic mixture, and each of these conditions was tested in duplicate. Thus, each supernatant was used on six wells, three conditions in duplicate.

  1. 1: which assay is used for viability. Not described here or in M&M. To give here OD does not make sense therefore.

Response: The Materials and Method section reports the following: “Sulforhodamine B colorimetric (SRB) assay was used to assess cell vitality. The method has been optimized to adherent cells in a 24-well plate. Cells were fixed with 50% trichloro acetic acid (TCA) at 4°C for 1h. Sequentially, the plate was washed 5 times with sterile water and left overnight to dry. The following day, cells were incubated with 300 ml of 0.4% SRB dissolved in 1% acetic acid, for 30 min in the dark; then the plate was washed 4 times with 200 ml of 1% acetic acid. After solubilization with TRIS 10 mM pH 10.5, absorbance was measured at λ = 540 nm by spectrophotometer (TECAN Spark, Milan, Italy).The absorbance was directly proportional to the protein content and therefore to the number of live cells present in each well.Caco-2 vitality was assessed after 6 h of incubation with the two concentrations of PF or only with DMEM served as blank (control)”. OD is the absorbance measured at 540 nm using a spectrophotometer.

  1. Results, 2,3: …recovered paracellular permeability… is wrong. Since bacteria and supernatant are given at the same time the effect is avoided or inhibited since it is not induced at the first place. This statement is also found at other places.

Response: We have replaced “recovered” with “avoided” throughout the manuscript.

  1. Results: paragraph 2.3 is found twice.

Response: We thank the reviewer for pointing out this discrepancy. We did not include the paragraph number in our submitted version, it was introduced by the journal and it is clearly an oversight. We have corrected this in the revised version of the manuscript.

  1. How is concentration of FITC calculated?

Response: As reported in the Materials and Methods section, a standard curve was used to extrapolate concentrations: “A standard curve was used to convert absorbance results in concentrations of FITC”.

  1. Why was FITC diffusion used instead of TEER.

Response: The choice of using FITC to assess changes in paracellular permeability is related to two reasons: 1) the measurement of TEER is the cumulative result of paracellular and transcellular resistance, whereas FITC provides information on paracellular permeability (the objective of the study) (PMID: 34513903); in co-incubation experiments with bacteria, the use of the TEER implies that the electrode of the voltmeter comes into contact with the bacteria, creating a problem of cleaning the electrode for subsequent re-use, as well as possible interference in the results (on the other hand, to exclude metabolization of FITC by the probiotics, we have previously incubated the two doses of Serobioma with the same amount of FITC used in the experiment and after 6 hours the amount of FITC had not changed).

  1. 5: what was measured should be shown in the panels.

Response: We have modified the figure by adding the name of the gene on the y-axis, as suggested by the reviewer.

Round 2

Reviewer 1 Report

Comments and Suggestions for Authors

REVIEW

Dear authors,

The recommended corrections were made, and some of the observations indicated in the Introduction and Materials sections were justified, which in my opinion makes the work easier to understand.

Please amend the requested comments and submit the revision file.

  1. In the title, the names of the microorganisms must be written in italics.

  1. The “mL” unit on the Y axis of the Figures must be corrected. 2A, 2B, 3 and 4A-F.

  1. In the References section, 21 to 30 are missing, please add them.

Please amend the requested comments and submit the revision file.

Author Response

Dear authors,

The recommended corrections were made, and some of the observations indicated in the Introduction and Materials sections were justified, which in my opinion makes the work easier to understand.

We thank the reviewer for his/her positive comment and for pointing out inconsistencies that we have corrected in the resubmitted version

Please amend the requested comments and submit the revision file.

  1. In the title, the names of the microorganisms must be written in italics.

Response: We thank the reviewer for this comment and apologize for the oversight. We have changed the manuscript accordingly. 

  1. The “mL” unit on the Y axis of the Figures must be corrected. 2A, 2B, 3 and 4A-F.

 Response: We thank the reviewer for this comment and apologize for the oversight. We have changed the figures accordingly. 

  1. In the References section, 21 to 30 are missing, please add them.

Response: We have added the missing references.

Reviewer 2 Report

Comments and Suggestions for Authors

The manuscript has sigificantly improved and could be published. Some very minor point need to be adressed.

English needs to be corrected at some places where due to corrections some errors were introduced.

In a few places still permeability was used instead of paracellular permeability. Please correct. E.g. 2.2 second paragraph. 2.4 first paragraph. Discussion one but last paragraph.

Comments on the Quality of English Language

see above

Author Response

The manuscript has sigificantly improved and could be published. Some very minor point need to be adressed.

We thank the reviewer for his/her positive comment.

English needs to be corrected at some places where due to corrections some errors were introduced.

Response: We have carefully read the manuscript and corrected the errors.

In a few places still permeability was used instead of paracellular permeability. Please correct. E.g. 2.2 second paragraph. 2.4 first paragraph. Discussion one but last paragraph.

Response: We thank the reviewer for this comment and apologize for the oversights. We have changed the manuscript accordingly.